# Selective Moonlighting Cell-Penetrating Peptides

**DOI:** 10.3390/pharmaceutics13081119

**Published:** 2021-07-22

**Authors:** Rafael Morán-Torres, David A. Castillo González, Maria Luisa Durán-Pastén, Beatriz Aguilar-Maldonado, Susana Castro-Obregón, Gabriel Del Rio

**Affiliations:** 1Department of Biochemistry and Structural Biology, Institute of Cellular Physiology, National Autonomous University of Mexico, UNAM, Mexico City 04510, Mexico; morantor@hu-berlin.de (R.M.-T.); dcastillo@ifc.unam.mx (D.A.C.G.); 2Laboratorio Nacional de Canalopatias, National Autonomous University of Mexico, UNAM, Mexico City 04510, Mexico; mlduran@ifc.unam.mx; 3Department of Neurodevelopment and Physiology, Institute of Cellular Physiology, National Autonomous University of Mexico, Mexico City 04510, Mexico; baguilar@ifc.unam.mx (B.A.-M.); scastro@ifc.unam.mx (S.C.-O.)

**Keywords:** cell-penetrating peptide, machine learning, moonlight protein, computational biology, protein function prediction, multifunctional protein

## Abstract

Cell penetrating peptides (CPPs) are molecules capable of passing through biological membranes. This capacity has been used to deliver impermeable molecules into cells, such as drugs and DNA probes, among others. However, the internalization of these peptides lacks specificity: CPPs internalize indistinctly on different cell types. Two major approaches have been described to address this problem: (i) targeting, in which a receptor-recognizing sequence is added to a CPP, and (ii) activation, where a non-active form of the CPP is activated once it interacts with cell target components. These strategies result in multifunctional peptides (i.e., penetrate and target recognition) that increase the CPP’s length, the cost of synthesis and the likelihood to be degraded or become antigenic. In this work we describe the use of machine-learning methods to design short selective CPP; the reduction in size is accomplished by embedding two or more activities within a single CPP domain, hence we referred to these as moonlighting CPPs. We provide experimental evidence that these designed moonlighting peptides penetrate selectively in targeted cells and discuss areas of opportunity to improve in the design of these peptides.

## 1. Introduction

Cell-penetrating peptides (CPPs) are a class of peptides that pass through biological membranes either by passive or active mechanisms [1,2,3]. This ability has promoted their use as carrier or delivery systems for basic [4,5] and applied [6,7] uses in biology and medicine, respectively.

An important limitation of these vectors is a relative lack of specificity, that is, CPPs may deliver their cargo inside many different cell types without distinction. While some CPPs are internalized through a receptor-mediated endocytosis mechanism, implying that such CPPs are specific for the cells expressing those receptors, most common CPPs are not selective (e.g., Tat, Octa-Arginine and penetratin). This has promoted the development over the past years of different strategies to provide selectivity to CPPs, namely: activation [8] and targeting [9,10]. The first strategy aims at switching CPPs from a “dormant” to an active state by different means (e.g., change in pH, temperature and enzymatic reaction) aiming at activating the CPP activity only at a certain cellular location or tissues; the targeting strategy adds a moiety/ligand to a CPP that recognizes a cell-surface molecule, hence directing the CPP towards the internalization into those cells. In either case, CPPs may significantly increase their molecular mass to incorporate selectivity, increasing the cost of synthesis and/or the chances to be recognized by the immune system. Furthermore, in our experience some combinations of ligands to CPPs may render peptides inactivation presumably by the interaction between the ligand and the CPP (data not published).

A possible solution to these problems is to design short selective CPPs; the reduction in size is accomplished by embedding two or more activities within a single CPP domain, referred to as moonlighting peptides [11], in this case, CPPs embedding ligands or activation peptide sequences. Since the added activities do not reside in different domains, but are included within the CPP activity, the chance to blocking each other is eliminated and the size of the resulting peptide is reduced. We recently described a computational approach based on machine learning to design moonlighting CPPs [12]; we showed then that with our computational strategy it is possible to transform a peptide sequence into a CPP by adding at least four residues. In the present work, we used the aforementioned computational approach to design two groups of moonlighting CPPs: (i) one group was designed based on the activation strategy (tested on mammalian cells) and (ii) another one based on the targeting strategy (tested on yeast cells). Testing our designs on these two experimental systems has several goals, for instance, to show the usefulness of our computational designs in mammalian and microbial cells, hence opening the possibilities to use this computational strategy to label or target these two types of cells. In both cases, the CPP activity was achieved by adding 4 or up to 12 residues, depending on the starting sequence. For mammalian cells we used the neprilysin substrate sequence for activating the CPP activity, while for yeast cells we used the α-pheromone to target the MatA cells. We provide experimental evidence that our designed moonlighting CPPs indeed display selective penetration activity, particularly in yeast cells. Furthermore, none of our CPPs were inactive. In this study we focus on testing the specificity of our designs and did not study the mechanism of internalization, because the mechanism of internalization should depend on the CPP used and the concentration used of the peptide. We discuss different areas of opportunities to apply this strategy and areas for future development.

## 2. Materials and Methods

### 2.1. Materials

The list of peptides used in this study is presented in Table 1. Please note all the peptides in this study are linear.

All these peptides were chemically synthesized by GenScript and purified >95%; the peptide sequences were validated by the peptide mass. All these peptides were modified to include in the N-terminus the fluorophore 5(6)-carboxytetramethyl rhodamine (TAMRA). The cell lines used in this study are presented in Table 2.

### 2.2. Machine Learning

A dataset of examples of the class to be learned is required to train a machine-learning algorithm. In the case of the targeting strategy, these examples are different peptide sequences represented by molecular descriptors (see below) with known CPP activity, while the negative dataset includes peptide sequences that do not share features commonly found in CPPs. These data sets are available from github: https://github.com/cdiener/dcf/tree/master/examples (accessed on 21 July 2021). In the case of the activation strategy, the examples are the non-CPPs and the negative dataset includes CPPs. A total of 27 attributes derivable from peptide sequences using the software provided at https://github.com/cdiener/dcf (accessed on 21 July 2021) were calculated using the predict program. The 27 attributes computed by this code include: (a) the counts for each of the 20 amino acids in a given protein sequence; (b) hydrophobicity; (c) mean charge; (d) isoelectric point; (e) sliding window range of charge; (f) sliding window range of the hydrophobic moment; (g) water-octanol partition coefficient and (h) an approximation of the alpha-helical content in the sequence. The random forest algorithm was implemented and provided for users to train different datasets. Briefly, this algorithm grows many classification trees, therefore the term “forest” was used. For each tree, a random subset of examples from the training data was retrieved. Then, a random number of the 27 attributes were selected. Based on this data, the algorithm built a random forest tree. Since each tree had a unique classification pattern, a consensus tree was derived from the average of all the trees. Please refer to our website (https://github.com/cdiener/dcf, accessed on 21 July 2021) for instructions on how to compile and execute this program. For the current work, the targeting strategy used the α-pheromone from *S. cerevisiae* as the ligand to direct the CPPs towards the yeast cells expressing the receptor for this pheromone (MATa). The α-pheromone was embedded within a CPP sequence by adding residues at both the N- and C-terminus regions of the pheromone; for further details of these designs please refer to our previous work on [12]. For the activation strategy, the Tat peptide sequence was used as a CPP and was embedded within a non-CPP sequence by adding to its C-terminus region the neprilysin recognition sequence (FFGFLA; neprilysin cuts after the G) and residues required to render a non-CPP peptide (DDDDEE).

### 2.3. Quantifying the Internalization in Saccharomyces cerevisiae Cells

We adapted a previous protocol reported to analyze fluorescent single yeast cells [13]. Briefly, cells were grown overnight and diluted to a final optical density measured at 600 nm of 1.0. A total of 116 μL of this cell dilution was mixed with 4 μL of a solution of penetrating peptide at 200 mM in a 1.5 mL plastic tube. This mixture was analyzed every 5 min for 60 min in a flux cytometer coupled to a multiphotonic microscope (Amnis ImageStream Mark II). In that equipment, 10,000 pictures were taken to record green (excitation 488 nm and emission 509 nm) and red (excitation 561 nm and emission 669 nm) fluorescence and the bright field. These images were then analyzed with the IDEAS software that is part of the Amnis equipment, to identify peptide internalization (red fluorescence). Two types of experiments were conducted; in one set, we exposed MATa and MATα cells separately to peptides α-NLS-C, NLS-α-CE and chimera (see Table 1 for a description of the peptide sequences). In another set of experiments, we exposed MATa and MATα cells mixed together to different penetrating peptides labeled with TAMRA; for this last experiment, the ribosomal protein RPL9A was fused to GFP in MATa cells, to facilitate their distinction from MATα cells. Events recorded in this way were further analyzed to discard cell debris and images that included clumps of cells. Images with a total area of 20–50 μm and an aspect ratio of 0.7 allowed us to identify events that include 1 or 2 cells; furthermore, images with less than 1500 arbitrary units of fluorescence were discarded. The data were plotted using ggplot2 from the R package. Please note that the reported experimental conditions were selected among several trials on the AMNIs equipment to identify the best conditions for these experiments.

To confirm that the fluorescence detected using the AMNIS multiphotonic microscope was the fluorescence inside cells, we carried out confocal microscopy experiments (Zeiss LSM510 META) by adapting a previous report [11]. For that end, yeast cells for each yeast haplotype were grown overnight for 14 h in YPD (1% yeast extract (USB Corp., Cleveland, OH, USA), 2% bacto peptone (Becton, Dickinson and Company, Franklin Lakes, NJ, USA) and 2% glucose (Merck, Naucalpan de Juárez Edo. De Mexico, Mexico). Cells were then centrifuged and resuspended in YPD. An OD600 of 0.4 was used for the confocal microscopy experiments, whereas for the cytometry experiments (above) a value of 1.0 was chosen. Yeast cells were immobilized using low melting point agarose (Agarose SFR VWR Life Technologies, Radnor PA, USA) at 0.5%. The peptide concentration used for these experiments was 50 μM. Prior to performing any observations, 3 cycles of sonications (sonicator Bransonic 32) for 3 min each were performed to dissolve potential peptide aggregates. A total of 12.5 μL of yeast inoculum, 1.5 μL of TAMRA-α-NLS-C solution and 1 μL of previously heated agarose were placed on a microscope slide. The process to collect images from each sample started immediately after the addition of the agarose. After identifying a field with immobile cells, pictures were taken every 3 min for 1 h in total, using a z-stack of 10 levels. Images were analyzed using Fiji software and the fluorescence intensity recorded for each cell was divided by the area of the corresponding cell.

### 2.4. Quantifying the Internalization in HEK293T, HEK293T-NEP and HeLa Cells

#### 2.4.1. Cell Culture

All cell lines were maintained as previously described [14] in DMEM (Gibco, Waltham, MA, USA) containing 10% FBS (ThermoFisher-Gibco, Waltham, MA, USA), 5000 U penicillin/streptomycin (ThermoFisher-Invitrogen, Waltham, MA, USA) and incubated at 37 °C with 5% CO_2_. Cells were seeded at a density of 3 × 10^5^ cells/well in a 6-well culture plate for Western-blot experiments and 4 × 10^5^ cells/well in 12-well culture plate for peptide internalization assays.

#### 2.4.2. HEK293T-NEP Stable Transfection

HEK293T cells were transfected as previously described [14] with the plasmid PCSC-SP-PW-Nep (Addgene, Watertown, MA, USA), which contains the human neprilysin (MME) gene. This transfection was done with the use of the reagent polyethylenimine (PEI) according to the stable transfection procedures for adherent cells. The transfected HEK293T cells were cultured with DMEM containing 10% FBS and Zeocin 400 µg/mL (ThermoFisher, Waltham, MA, USA) to select stably transfected cells, which were recovered after the third day of transfection. The selected resistant cells were cultured under the Zeocin selection medium for approximately 2 months, then the HEK293T-NEP cells were maintained in the culture medium containing 300 µg/mL of zeocin; this concentration allowed the growth of the transfected cells, but not the wild-type. The cell line transfected with PCSC-SP-PW-Nep was designated as HEK293T-NEP.

#### 2.4.3. Western-Blot

Immunoblot was performed following the standard procedure [15]. Cells were homogenized in a lysis buffer (60 mM Tris (pH 6.8), SDS 2% and 2 mg/mL protease inhibitor complete (Merck, Naucalpan de Juárez Edo. de Mexico, Mexico) at room temperature for 15 min. The 6-well dishes were scraped, sonicated and centrifuged for 10 min at 13,000× *g* to remove cell debris. A total of 50 μg of total protein were separated by SDS/PAGE in 10% gels. Proteins were electrotransferred to the PVDF-FL membrane (Millipore, Germany) and incubated overnight at 4 °C with the anti-CD10 (ab79432 Abcam, USA; diluted 1:5000) primary antibody. After three washes with TBST, membranes were incubated for 1 h with anti-rabbit IRDye800 (LI-COR, Mexico city, Mexico; diluted 1:5000) secondary antibody. Membranes were scanned in an IRDYE 800CW image system and protein quantification was inferred by image analysis using Odyssey Image Studio software 5.2.5 (LI-COR, Mexico City, Mexico).

#### 2.4.4. Internalization Assays

This procedure was adapted from a previously reported protocol [13]. Cells were incubated with TatNep, TatNepNoCPP or D-TatNepNoCPP peptides at a concentration of 10, 5 and 1 μM (see Table 1). All peptides were dissolved in a 1:4 DMSO-Water solution. Cells were incubated with each peptide for 30 min, then washed with PBS three times and fixed with 4% paraformaldehyde (Merck, Naucalpan de Juárez Edo. de Mexico, Mexico) for 30 min at room temperature. Afterwards, samples were washed with PBS three times and then incubated with 1 μg/mL of DAPI for 2 min at room temperature. For the images, 1000 cells were taken (3 replicates) by using the epifluorescent microscopy (Nikon ECLIPSE Ti-U). Due to a slight precipitation of the peptides, to avoid interference with the quantification, the 10% with the highest fluorescence intensity were considered outliers and were removed for all conditions. A total of 2700 cells remained for further analysis. Internalization analysis was performed by confocal microscopy. Cells were incubated with any of the three peptides at a concentration of 5 μM for 30 min (the fluorescence was strong and the precipitation reduced), then cell membranes were stained with the CellMask^TM^ Green Plasma Membrane Stain (ThermoFisher, Waltham, MA, USA) following specifications of the provider. Images were acquired with a Zeiss LSM800 microscope using the ZEN 2.6 (blue edition) software. All images were analyzed with the Fiji/ImageJ Software and are available at http://bis.ifc.unam.mx/ironbios/selmoonCPP/ (accessed on 21 July 2021). To identify the statistically equal distribution of peptide internalization, we conducted a non-parametric Kruskal–Wallis test as implemented in python; pairs of experiments with *p* < 0.05 were considered significantly equal.

#### 2.4.5. Cytotoxicity Assays

Cells were harvested in 96-well plate dishes at a concentration of 2000 cells per well. All cell lines were incubated with TatNep, TatNepNoCPP or D-TatNepNoCPP peptides at a concentration of 10 μM for a period of time of 4, 8, 12 and 24 h. Calcein AM (ThermoFisher, Waltham, MA, USA) and DAPI (ThermoFisher, Waltham, MA, USA) were added to stain cells. Exposing cells to Triton X-100 0.1% and PBS, respectively, included dead and live cells controls. DMSO toxicity assays were conducted by using a 25% DMSO solution and added to cell cultures containing 2000 cells per well; the amount of DMSO used in these toxicity tests reproduced the amount of DMSO added in the experiments using 10 μM of peptides solubilized in DMSO, which was the largest amount of DMSO used in our experiments. Images were taken with the automatized epifluorescence microscope ImageXpress^®^ MicroXLS (Molecular Devices). The number of cells per well (live/dead) was analyzed with the Live/dead analysis module of MetaXpress^®^ program (Molecular Devices).

#### 2.4.6. Data Analysis

For the mammalian cells images obtained from epifluorescence microscopy, a normality Shapiro–Wilk test was done for each individual condition (27 conditions). None of the conditions showed a normal distribution (alpha = 0.05). To identify a statistically equal distribution of peptide internalization a non-parametric Kruskal–Wallis test was implemented; pairs of experiments with *p* < 0.05 were considered significantly equal.

The Western blots were analyzed considering the differences in number of experiments done for each cell line (HEK293T and HeLa cells 10 data, HEK293T-NEP 6 data) and the experimental differences noted between the gels. The density readings were normalized (unit L2 normalization) and the statistical differences between the data was performed using the Kruskal–Wallis test with an α = 0.05; hence experiments with *p* < 0.05 were considered different.

## 3. Results

### 3.1. Designing Moonlighting CPPs

Three targeted moonlighting CPPs previously designed in our group were tested for their selectivity to penetrate MATa cells. Since all peptides include the α-pheromone sequence embedded within a CPP sequence, these were previously shown to activate the pheromone receptor signaling, and should bind to the α-pheromone receptor (Ste2p) [12]; the sequences of these peptides are reported in Table 1 (α-NLS-C, NLS-α-CE and chimera) and the affinity constant (Kd) of the pheromone receptor for the α-pheromone varies from 2 [16] to 10 nM [17]. The design in that previous work required the training of a classifier to identify CPPs from non-CPPs sequences (see Methods). These three peptides presented a high probability to comply with CPP rules (P(CPP) >= 0.88), and we have previously shown in the aforementioned publication that all these peptides penetrate mammalian cells. In the present study their selectivity to penetrate yeast cells that express or not express the Ste2p receptor was tested (*S. cerevisiae* MATa cells; see Figure 1A).

Additionally, three other peptides were designed to test the activation strategy (see Table 1). In this strategy, a known CPP sequence (Tat) was embedded within a non-CPP sequence that included a protease cleavage site; such a design will cleave off the CPP at the site where the protease is expressed, hence activating its penetrating activity in the proximity of the target cells. For this strategy we chose the peptidase neprilysin (also known as CD10), known to be overexpressed in some types of cancer tissues [18]. The cleavage site sequence recognized by this protease has been well characterized [19] allowing us to add it to the C-terminal of the well-known CPP Tat (47–57) sequence; the reported Km for this sequence by neprilysin is 0.9 mM [19]. This peptide was optimized to become a non-CPP by adding six amino acid residues (see Table 1, TatNepNoCPP). The design in this case required the training of a classifier to identify non-CPPs from CPPs (see Section 2). This peptide would not be expected to penetrate cells, unless digested by neprilysin (see Figure 1B), yet the peptide should be recognized by NEP and consequently may be attracted to cells expressing this protease. As controls, we synthesized an all-D isomer of the aforementioned peptide, which should be neither digested by neprilysin nor recognized by cells expressing NEP (see Table 1, D-TatNepNoCPP), and the peptide produced by the digestion of TatNepNoCPP by neprilysin (see Table 1, TatNep) as a positive control for internalization. All these peptides were synthesized and coupled at the N-terminus to the fluorophore TAMRA to detect the internalization of the peptides. We have previously shown that yeast or mammalian cells do not internalize TAMRA unless fused to a CPP [12].

### 3.2. Experimental Testing of Targeted Moonlighting CPPs

It has been argued that cell fixation induces artificial internalization of peptides and that the use of flow cytometry alone cannot differentiate between peptides bound to the membrane from peptides inside cells [1]. The use of multiphoton microscopy coupled to flow cytometry applied to living cells reduces these problems (see Methods). Our protocol systematically took pictures of cells both by fluorescence or bright light; the analysis of these images allowed us to distinguish cells that are fluorescent at the periphery (peptides bound to the cell membrane) from cells that are fluorescent from within (peptides that are internalized). To test for the selectivity of pheromone-receptor-targeted peptides, living cells expressing the α-pheromone receptor (*S. cerevisiae* MATa cells) and living cells not expressing the receptor (*S. cerevisiae* MATα cells) were exposed to these peptides and the fluorescence intensity inside these cells was quantified every 5 min during 1 h (see Figure 2). The data is presented in box plots that summarize all times recorded. We observed that MATa cells exposed to the α-NLS-C peptide presented more selectivity towards MATα cells than NLS-α-CE and chimera peptides. MATa cells preferentially internalized all three peptides. In summary, the three designed peptides showed selectivity for cells that expressed the receptor for the ligand embedded within the moonlighting CPP.

To further explore the selectivity of these peptides, we mixed MATa:RPL9A-GFP and MATα cells in a single culture and exposed them to each of these three different peptides. MATa cells in this case expressed the ribosomal fusion protein RPL9A-GFP, to differentiate them from MATα cells. We observed that the three peptides selectively internalized into MATa (see Figure 3) cells with similar selectivity as observed when MATa and MATα cells were separately exposed to these peptides. We also observed that these data were less dispersed than when cells were exposed to the peptides separately.

Finally, we used confocal microscopy to locate TAMRA fluorescence from α-NLS-C peptide inside cells to further validate our previous observations. We observed that cells expressing the α-pheromone receptor STE2 (MATa cells) internalized more fluorescence than cells that did not express the receptor (see Figure 4). Together these results indicate that these peptides are selectively internalized into cells that express the receptor for the α-pheromone.

### 3.3. Experimental Testing of Activatable Moonlighting CPPs

To study the selectivity of the activatable cell penetrating peptides (ACPPs), we used HEK293T due to its extremely low level of neprilysin (NEP) transcripts as reported by the Human Protein Atlas [20] and HEK293T transfected cells stably overexpressing NEP (HEK293T-NEP); for comparison we also studied HeLa cells that express NEP. Expression differences on neprilysin between these cell lines were verified by Western blot. Our results indicate that HeLa and HEK293T-NEP cells expressed more neprilysin than HEK293T, respectively (see Figure 5), yet HEK293T cells did express neprilysin.

As a first approximation, internalization of the peptides was inferred from epifluorescence microscopy (see Section 2; the actual data is found in the Appendix A). The representation of this data is shown as box plots to facilitate the visual comparison in the internalization of each peptide on the three different cell lines described above (see Figure 6A). The positive control TatNep peptide (this peptide is a CPP) at 10 and 5 μM was internalized in HEK293T and the over-expressing HEK293T-NEP cells with the same efficiency (compare the medians in the figure) as expected; however, HeLa cells internalized more of this peptide at 10 μM than the other cell lines. The negative control D-TatNepNoCPP peptide was less internalized in all cell lines and is the only one that does not present a concentration dependent internalization (Figure 6A). Additionally, it is noticeable that TatNepNoCPP presents the best internalization with respect to the other peptides for any cell line (compare the Y-axis values presented in Figure 6A). To clarify this difference in the internalization of these peptides, we present in Figure 6B the three peptides clustered by the cell line; the median values at 10 μM for the data in Figure 6B are included in Appendix A.

Confocal microscopy was performed to validate that the fluorescence detected was indeed inside the cells (see Section 2). Interestingly, three different patterns of fluorescence were observed (see Figure 7): (i) diffused fluorescence (blue arrow), (ii) punctuated fluorescence (yellow arrow) and (iii) a combination of these two (white arrow). TatNep is more often seen as a punctuated fluorescence while TatNepNoCPP and D-TatNepNoCPP as a diffused layer covering the cells. The orthogonal representation shows that all three peptides were detected inside cells, yet cells exposed to D-TatNepNoCPP displayed less internalization compared with the other two peptides in all three cell lines (see Figure 7). No toxicity was observed during these experiments or during an experimental quantification of toxicity (see Appendix A), hence supporting the notion that these peptides are internalized through the Tat peptide once it is activated.

## 4. Discussion

Cell-penetrating peptides represent a powerful tool to deliver biologically active compounds inside cells that otherwise cannot reach their intracellular targets. An important limitation of CPPs is the lack of specificity, but two different approaches have been proposed to overcome this limitation: add a ligand to target a surface protein or activate the CPPs until a condition is met (e.g., protease activity). Here we argue that any of these approaches has inherent limitations due to the increase not only of the total size of the peptide mass, but also the cost of synthesis, susceptibility to proteases and immunogenicity. To overcome these inherent limitations in targeting CPPs, we propose to embed within the CPP sequence any other activity to create a single-domain multifunctional peptide that we refer to as selective moonlighting CPPs. Using our previously described computational approach to design multifunctional single-domain (moonlight) CPPs, here we tested this approach in rendering two classes of selective CPPs: activatable and targeted CPPs. Our results show that both approaches generate selective CPPs against the desired cells: targeted CPPs penetrate selective cells expressing the receptor for the ligand embedded into the CPPs and activatable CPP penetrated more than the hydrolyzed CPP product in cells expressing the protease that activates the CPPs. However, we noted some important details that deserve further discussion in order to plan for future improvements.

In the targeted strategy, we noticed differences in penetrating efficiencies among the three targeted CPPs. In particular, NLS-α-CE and chimera peptides penetrate less efficiently MATa cells expressing the targeted surface protein (Ste2p) than the α-NLS-C peptide. This is observed when MATa are exposed to these peptides separately or mixed with MATα cells. We have previously noted that NLS-α-CE and chimera peptides, but not a α-NLS-C peptide, are toxic to MATa cells [12]. However, if NLS-α-CE and chimera peptides would be disrupting the membrane potential of MATa cells as part of their toxicity, we would expect these peptides to penetrate more easily these cells than the α-NLS-C peptide, which is the opposite of what we observed, hence the observed reduced internalization of NLS-α-CE and chimera peptides may not be explained by the toxicity reported for these peptides. On the other hand, we have reported that modifying the C-terminus region of the pheromone reduces its activity in a moonlighting peptide [13]. The NLS-α-CE is the only peptide that did not modify the C-terminus region of the pheromone and consequently we would not expect this peptide to reduce its affinity for its receptor, while the α-NLS-C and chimera peptides did, providing a possible explanation for their reduced ability to selectively penetrate MATa cells. Alternatively, some of the added residues may sometimes compensate for the possible deleterious effect of adding residues to a peptide functional domain; such mutations are referred to as compensatory mutations. Thus, we anticipate that the design of moonlighting peptides needs to consider compensatory mutations in the added residues, a feature not currently implemented in our computational approach. While compensatory mutations may be inferred from multiple sequence alignments of naturally occurring proteins [21,22,23], to the best of our knowledge, there are no currently available computational methods to predict compensatory mutations in the absence of homologous proteins and consequently these methods cannot be applied to computer-generated peptide sequences. This represents an area of opportunity.

ACPPs were first proposed in 2004 by Jiang [24] where a polyarginine-based CPP was fused to an inhibiting negatively charged domain via a linker peptide sequence cleavable by matrix metalloproteinases (MMPs) 2 and 9. Shi [25] and Yoo [26] used these same enzymes in later works, showing that cell lines over-secreting MMPs (HT-1080 cells) preferentially internalized ACPPs or when the proteases have been added to the media. We have previously shown that ligands embedded into CPPs may use receptors to promote their internalization by a mechanism independent of endocytosis [27]. Here we noted that the use of membrane-associated metalloendopeptidase (MME) may function as a receptor for the ACPP, hence attracting the peptide to the protease-expressing cells; such an attraction may promote the internalization of ACPPs that have not been cleaved/activated. For instance, we observed that the ACPP including the cleavage sequence recognized by neprilysin (TatNepNoCPP) was associated to target cells more than the CPP without the full-length peptide sequence recognized by neprilysin (TatNep) for all cell lines. The differences in the internalization between peptides per cell line is shown in Figure 6B, where it follows the order TatNepNoCPP > TatNep > D-TatNepNoCPP for all cell lines. The ratios between the observed medians in this table can be found in Appendix A. For simplicity, here we compared the average of the ratios between the peptides per cell line at the maximum concentration of 10 μM. We observe that the internalization of TatNepNoCPP was 3 times higher than TatNep and 8.8 times higher than D-TatNepNoCPP and this last peptide was 3 times lower than TatNep.

However, we did not observe higher internalization of TatNepNoCPP in oversecreting cell lines (HEK293T-NEP) than in non-oversecreting cell lines (HEK293T and HeLa), instead we observed the opposite at a concentration of 10 and 5 μM where the internalization preference was HeLa > HEK293T > HEK293T-NEP (Figure 6A). A possible explanation for this apparent discrepancy is the presence of post-translational modifications (glycosylations) that are required to activate neprilysin [28,29]. Thus, the difference in the internalization of TatNepNoCPP between these cell lines can be affected by the difference in the glycosylation levels of the protease on each cell line. Further experiments are needed to verify this.

Interestingly, HeLa cells present the highest internalization median for all peptides at 10 μM. Mueller [30] reported similar results by comparing the uptake of 22 CPPs in non-trypsinized HEK293 and HeLa cells. Such a difference in internalization depends on cellular membrane lipid composition, which may alter in turn membrane fluidity, proteoglycans and sphingomyelin content [31,32] and consequently differences in internalization efficiency may be expected depending on the chosen target cell. As Muller and collaborators noted, this preference to internalize by HeLa cells may be reversed by trypsinizing these cells. The non-trypsinized experiments are closer to the conditions cells will have in a tissue, therefore this is more adequate to anticipate results of targeted CPPs in animal studies.

All the activatable moonlighting CPPs displayed specificity. However, these presented poor solubility in water and we have to use dimethyl sulfoxide (DMSO) and water to dissolve them (see Section 2). While DMSO is widely used to solubilize a variety of chemical compounds, it is also known to reduce cell viability [33]. Thus, it is convenient to work with soluble moonlighting peptides. Among the molecular descriptors used to describe peptides, we included both the hydrophobicity and water-octanol partition coefficient (see Section 2); this is because CPPs are expected to display the affinity for membranes and in consequence these would be expected to be insoluble in water. On the other hand, CPPs tend to be positively charged and these may compensate for their low solubility in water. As shown in the case of the targeted moonlighting peptides, all these peptides were soluble in water and penetrated cells, hence it is possible to maintain the ability to penetrate cells with the ability to be soluble in water. This indicates that other chemical descriptors need to be explored to predict CPP solubility in water. Recent advances in making membrane proteins soluble in water [34] may provide some elements to improve on the design of water-soluble CPPs.

Targeted CPPs may have multiple applications. For instance, these peptides may be used to target drug action into specific cell types in cell cultures [35,36] or in animals [10,37]. Multiple studies aimed at studying cell physiology would benefit from having access to such peptides; for instance studies on yeast mating [38] or neuroscience [39]. We expect that our work may inspire other groups to use our computer-based designs in future studies.

In summary, we described the use of a computational strategy to design selective moonlighting cell-penetrating peptides based on two different mechanisms, namely targeted and activatable CPPs. Experimental evidence is presented that validates the specificity of these peptides. These moonlighting CPPs solve the current limitations of previously proposed approaches to render selective CPPs. We propose two possible areas to improve on the current computational approach used to generate moonlighting CPPs, including prediction of compensatory mutations and of peptide sequences that are soluble in water.

## Figures and Tables

**Figure 1 pharmaceutics-13-01119-f001:**
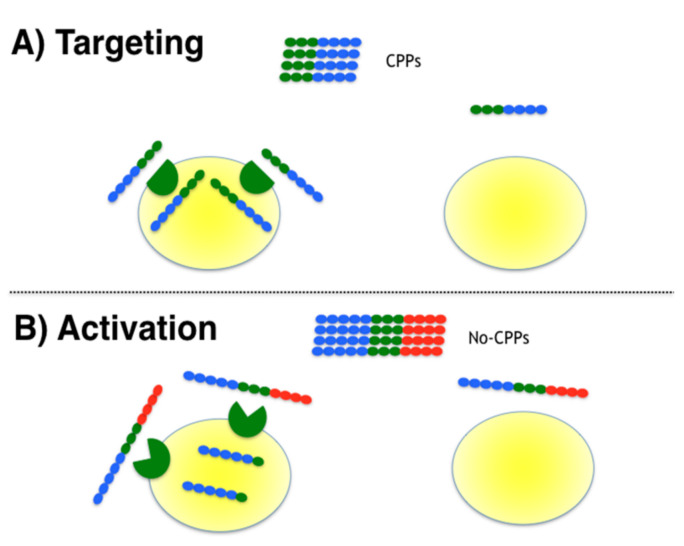
Strategies to provide selectivity to CPPs. Two strategies were tested to provide selectivity to CPPs, namely targeting and activation. The CPP sequence in blue, non-CPP conferring function sequence in red and ligand and cleavage site in green for (**A**) targeting and (**B**) activation, respectively.

**Figure 2 pharmaceutics-13-01119-f002:**
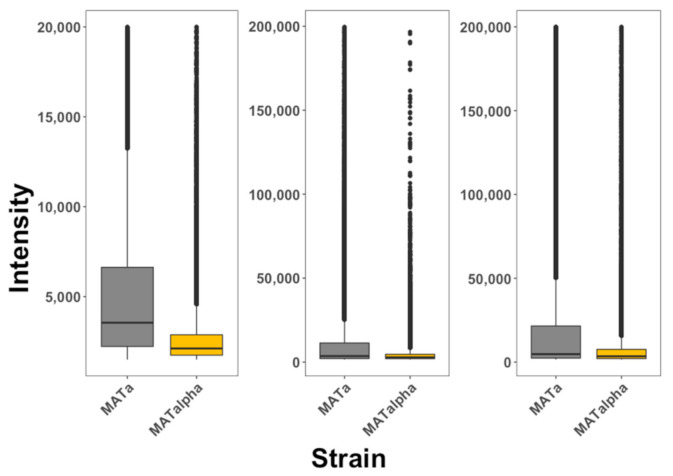
Internalization of selective CPPs on yeast cells. MATa (gray) and MATα (golden) cells were exposed to α-NLS-C (left panel), NLS-α-CE (middle panel) and chimera (right panel) peptides. The degree of internalization is proportional to fluorescence intensity. The data is presented as box plots, where the second and third quartiles are presented below and above the dark horizontal line representing the median. The data represented in this plot is included as Appendix A and corresponds with 10,000 cells analyzed for each condition. These experiments were repeated at least 3 times for each condition.

**Figure 3 pharmaceutics-13-01119-f003:**
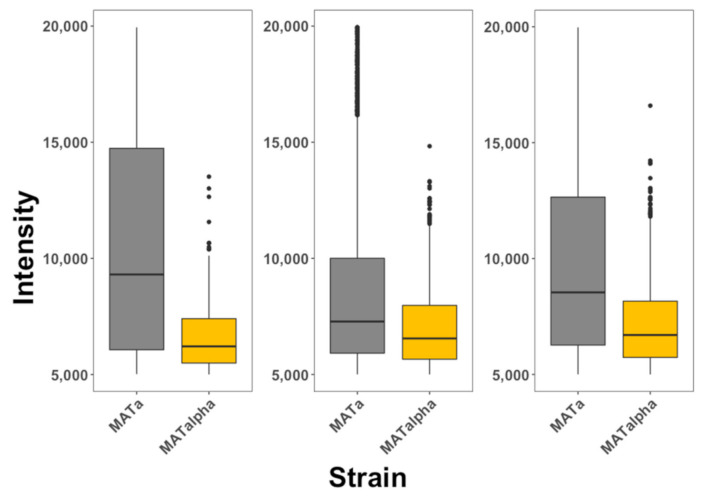
Internalization of selective CPPs on mixtures of yeast cells. MATa: RPL9A-GFP (gray) and MATα (golden) cells were combined in a single vessel and exposed to α-NLS-C (left panel), NLS-α-CE (middle panel) and chimera (right panel) peptides. The degree of internalization is proportional to fluorescence intensity. The data is presented as box plots, where the second and third quartiles are presented below and above the dark horizontal line representing the median. The data represented in this plot is included as Appendix A and corresponds with 10,000 cells analyzed for each condition. These experiments were repeated at least 3 times for each condition.

**Figure 4 pharmaceutics-13-01119-f004:**
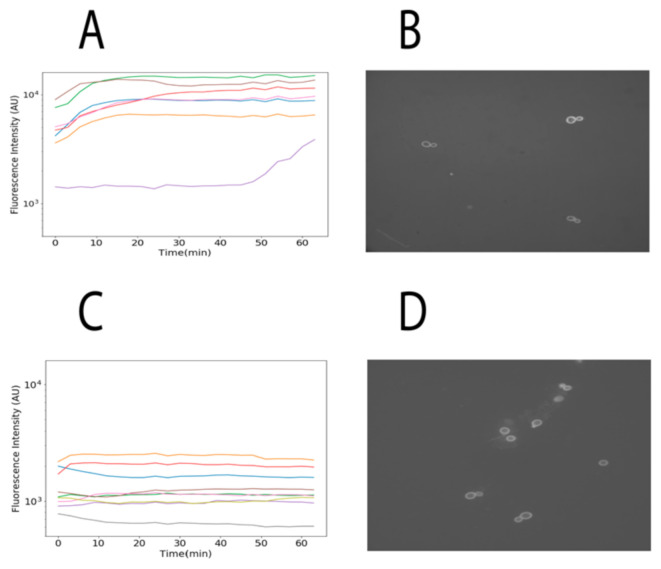
Internalization of α-NLS-C followed by confocal microscopy. (**A**,**C**) present the accumulated and normalized fluorescence intensities captured for each cell observed in the microscope for MATa and MATα, respectively. (**B**,**D**) present the cells MATa and MATα at the beginning of the experiment, respectively. Two sets of experiments were conducted; in this image we provide a summary of one of those experiments. The set of images captured each 3 min during 1 h are available at http://bis.ifc.unam.mx/ironbios/selmoonCPP/ (accessed on 21 July 2021) as Appendix A.

**Figure 5 pharmaceutics-13-01119-f005:**
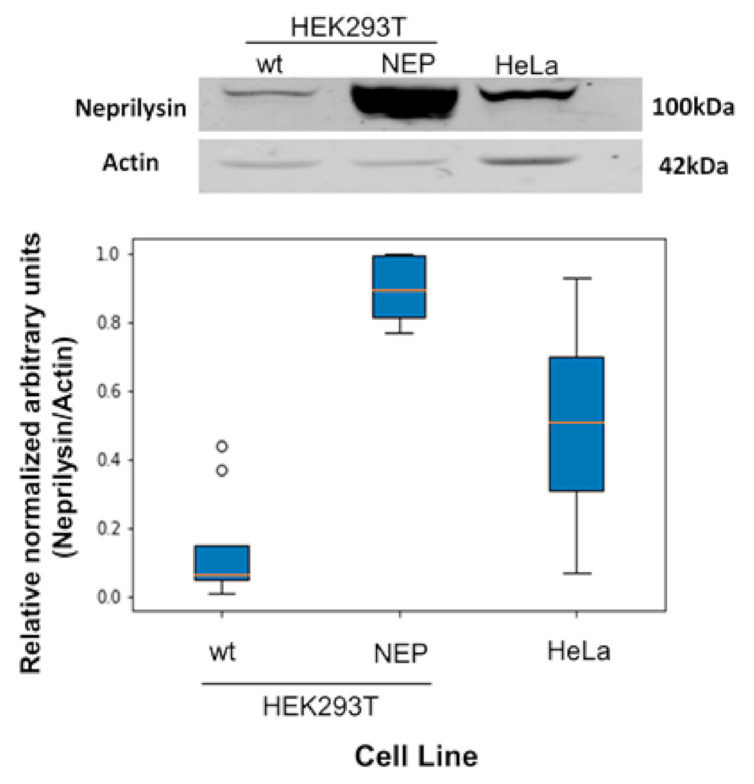
Protein expression levels of neprilysin in HEK293T, HEK293T-NEP and HeLa cell lines. The top panel shows a representative Western blot comparing NEP expression level in HEK293T cells parental (Wt) and HEK293T transfected cells stably over-expressing NEP (HEK293T-NEP) and HeLa cells. The bottom graph shows a densitometric analysis of the corresponding expression (*n* = 6 for HEK293T-NEP cells and *n* = 10 for the other cell lines). All paired comparisons reported *p* values < 0.05, hence all are considered significantly different. Raw images for the Western blots are presented as Appendix A.

**Figure 6 pharmaceutics-13-01119-f006:**
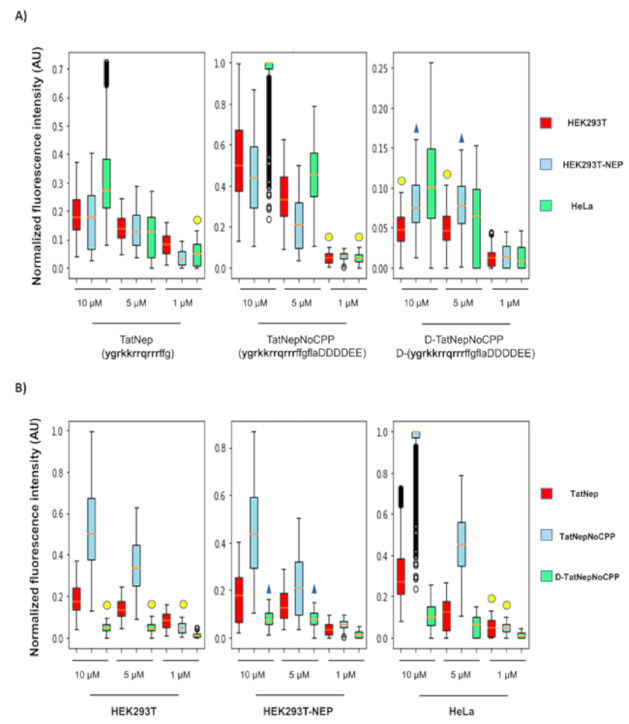
Internalization of activatable CPPs in mammalian cells. Normalized fluorescence intensity found in 2700 cells for each peptide. (**A**) Cell dependent internalization for HEK293T (red), HEK293T-NEP (blue) and HeLa (green). (**B**) Peptide dependent internalization for TatNep (red), TatNepNoCPP (blue) and D-TatNepNoCPP (green). Statistically similar distributions between the experiments are indicated with yellow circles or blue triangles (*p* < 0.5, see Methods). The data represented in these boxplots are included in Appendix A and as Appendix A at http://bis.ifc.unam.mx/ironbios/selmoonCPP (accessed on 21 July 2021). The data presented correspond with 1000 images per condition and 3 replicates.

**Figure 7 pharmaceutics-13-01119-f007:**
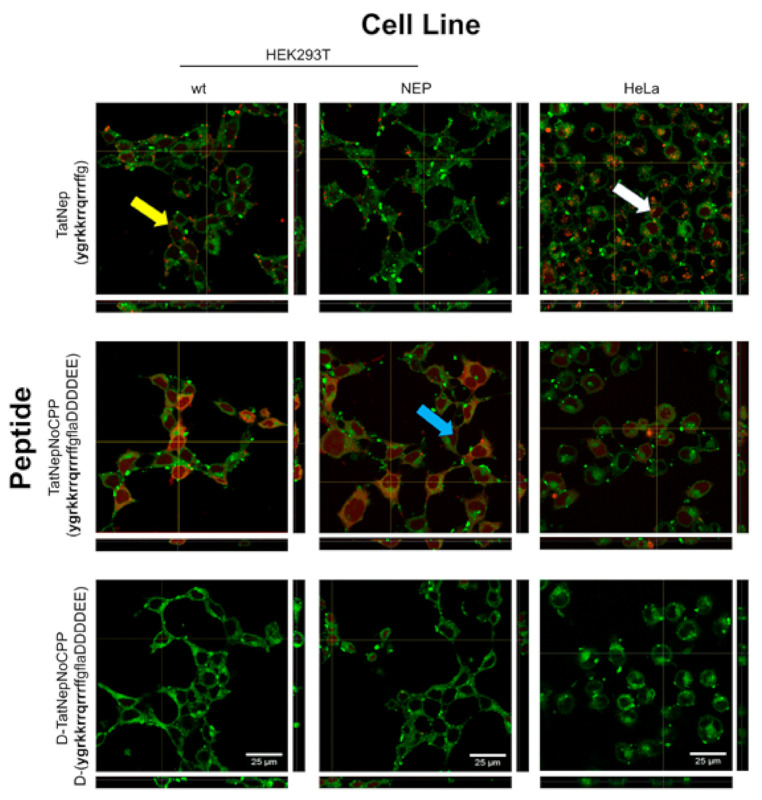
Confocal microscopy of mammalian cells exposed to selective CPPs. Orthogonal projections of cells exposed to fluorescent CPPs (5 μM) to estimate internalization in HEK293T, HEK293T-NEP and HeLa cell lines are presented. Original stack images (“xy” coordinates) in the middle and “yz” and “xz” projections at the right and bottom, respectively. Yellow (punctuated), blue (diffuse) and white (punctuated and diffuse) arrows indicate examples of the three distinct fluorescent patterns observed. White bars represent 25 μm.

**Table 1 pharmaceutics-13-01119-t001:** Moonlighting CPP sequences used in this study.

ID	Sequence	P(CPP)
α-NLS-C	RL**WHWLQLKPGQPMY**WRQPKSKRKVRR	0.91
NLS-α-CE	FRKWRRKPKKKRKVWWRKVKRR**WHWLQLKPGQPMY**	0.97
Chimera	**KRRWRFVWMNPKKKRKVPPWPYLLWWHWLQLKPGQPMY**	0.88
TatNep	**YGRKKRRQRRR**FFG	0.99
TatNepNoCPP	**YGRKKRRQRRR**FFGFLADDDDEE	0.112
D-TatNepNoCPP	D-(**YGRKKRRQRRR**FFGFLADDDDEE)	0.11

The name of the peptide (ID column), the peptide sequence and the calculated probability to match the CPP rules (P(CPP)). The α-pheromone sequence, the sequence recognized by neprilysin and the Tat sequence are indicated in bold; the plain letters represent the amino acids added to satisfy the rules of CPP (α-NLS-C, NLS-α-CE and chimera) or no CPP (TatNep, TatNepNoCPP and D-TatNepNoCPP).

**Table 2 pharmaceutics-13-01119-t002:** Cell lines used in this study.

Cell Line	Genotype	Provider
*S. cerevisiae* MATa	BY4741: his3D1; leu2D0; met15D0; ura3D0	*
*S. cerevisiae* MATα	BY4742: his3D1; leu2D0; lys2D0; ura3D0	Euroscarf
*S. cerevisiae* MATa: RPL9A-GFP	BY4741: his3D1; leu2D0; met15D0; ura3D0; RPL9A-GFP	*
*S. cerevisiae* MATa:*bar1*Δ	BY4741: his3D1; leu2D0; met15D0; ura3D0; *bar1*Δ	*
HEK293T	ND	#
HeLa	ND	#

* These strains were kindly provided by Dr. Alexander De Luna at LANGEBIO, Mexico. # These cell lines were kindly provided by Dr. Juan Carlos Gomora at UNAM. ND: Not determined.

## Data Availability

Supplementary data and codes reported in this work are available at: https://github.com/cdiener/dcf/ and http://bis.ifc.unam.mx/ironbios/selmoonCPP (accessed on 21 July 2021).

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
