# Peer review of "Selective Moonlighting Cell-Penetrating Peptides"

_pharmaceutics, 2021, doi:10.3390/pharmaceutics13081119_

Round 1
Reviewer 1 Report
In this paper, Del Rio and co-workers are testing some ideas to make CPPs cell specific. The work should be of interest to a broad community.
I have however several comments/concerns:
1) the use of yeast as a model is unusual and a justification for this choice is not provided. it is also unclear how results with yeast will relate to human cells. Images in Figure 4 describe "internalization" in yeast. However, it appears that the dominant signal is cell wall/plasma membrane association.
2) Penetration is loosely defined, and can be interpreted as endocytosis and/or cytosol delivery. The authors need to define specifically what is measured.
3) Figure 2. The figures are generally "stretched in the manuscript". Figure 2 is hard to read and the black line formed but all the data points not informative. The panel on the left had a max of 20000, the two panels on the right have a max of 200000. I am not sure if this is correct and if so, why the scales are different.
4) it is unclear why the peptides have lower-case letters, as this convention is usually used for D-amino acids.
5) The diffuse fluorescence in Figure 7 is interpreted as "cytoplasmic". On one hand, it is possible. On the other hand, these images are not sufficient to make this claim. In particular, TAT peptides typically show nucleolar staining upon reaching the cytoplasm/nuclei. The red fluorescence observed here could instead be out of focus cell sticking. Moreover, diffuse fluorescence can also be observed as an imaging artifact (e.g. photolysis of endosomes by the excitation light of TMR during imaging). Finally, fixing cells can also lead to endosomal escape and a diffuse signal (it is not clear to me if/how/when the 4% paraformaldehyde step is use in the imaging of Figure 7).
6) the authors need to show that the signal they measure is not cell sticking (given the precipitation concerns). If it is in endosomes, they should demonstrate that. The internalization assay also needs to assess that the cells used for quantification are not dead/dying. A final concern is that the TAMRA signal could be partially quenched in some conditions but not otehrs, hence biasing the results. Fluorescence measurements from cell lysates (where quenching would be less likely) should be done and compared to those obtained by microscopy.
Author Response
In this paper, Del Rio and co-workers are testing some ideas to make CPPs cell specific. The work should be of interest to a broad community.
I have however several comments/concerns:
1) the use of yeast as a model is unusual and a justification for this choice is not provided. it is also unclear how results with yeast will relate to human cells. Images in Figure 4 describe "internalization" in yeast. However, it appears that the dominant signal is cell wall/plasma membrane association.
We appreciate this note. Indeed, we did not emphasize the usefulness of the yeast cells. We have now added the following description at the introduction to clarify this:
In the present work, we used the aforementioned computational approach to design two groups of moonlighting CPPs; one group was designed based on the activation strategy (tested on mammalian cells) and the other one on the targeting strategy (tested on yeast cells); testing our designs on these 2 experimental systems has several goals, for instance, to show the usefulness of our computational designs in mammalian and microbial cells, hence opening the the possibilities to use this computational strategy to label or target these two types of cells.
The intensity recorded in the experiments summarized in Figure 4 are actually from confocal microscopy. Section 2.3 includes a description on how this internalization was detected, but we realized that we missed to explain how we focused on the internal part of the cells. To clarify this, we have now added the following description in Methods:
To confirm that the fluorescence detected using the AMNIS multiphotonic microscope was inside cells, we carried out confocal microscopy experiments (Zeiss LSM510 META), by adapting a previous report [11]. For that end, yeast cells for each yeast haplotype were grown overnight for 14 hours in YPD (1% yeast extract (USB Corp., USA), 2 % bacto peptone (Becton, Dickinson and Company, USA) and 2% glucose (Merck, USA)). Cells were then centrifuged, and resuspended in YPD. An OD600 of 0.4 was used for the confocal microscopy experiments, whereas for the cytometry experiments (above) a value of 1.0 was chosen. Yeast cells were immobilized using low melting point agarose (Agarose SFR VWR Life Technologies, USA) at 0.5%. Peptide concentration used for these experiments was 50 μM. Prior to performing any observations, 3 cycles of sonications (sonicator Bransonic 32) for 3 min each were performed to dissolve potential peptide aggregates. 12.5 μL of yeast inoculum, 1.5 μL of TAMRA-α-NLS-C solution and 1 μL of previously-heated agarose were placed on a microscope slide. The process to collect images from each sample started immediately after the addition of the agarose. After identifying a field with immobile cells, pictures were taken every 3 minutes for 1 hour in total, using a z-stack of 10 levels. Images were analyzed using Fiji software and the fluorescence intensity recorded for each cell was divided by the area of the corresponding cell.
Since the images presented in Figure 4A and 4C, correspond with the sum of all intensities recorded for each cell, we are now including as supplementary material (we added these set of images in a web site due to the large size of these images: http://bis.ifc.unam.mx/ironbios/selmoonCPP/) the set of images recorded for the experiments presented in Figure 4. In these set of images is clear the trend shown in Figure 4: the α-NLS-C peptide in inside MatA cells and not observed inside MatAlpha cells.
2) Penetration is loosely defined, and can be interpreted as endocytosis and/or cytosol delivery. The authors need to define specifically what is measured.
This is an interesting point. The goal of our work was to identify peptides that reside inside cells, yet the mechanism of internalization was not addressed in this work. We believe that aspect corresponds to another work. For instance, we have previously reported a methodology that precisely assists in the identification of the mechanism of internalization involved on CPP, which requires a combination of genetic, physical and computational strategies (https://pubmed.ncbi.nlm.nih.gov/28013505/; https://pubmed.ncbi.nlm.nih.gov/24706763/). Our current work focuses on how to target CPP action onto specific cell types by two different strategies: activation and targeting. The internalization mechanism should depend on the particular CPP used and the concentration of the peptide, which in turn depends on the particular use any of these peptides are aimed to. For instance, targeting yeast cells of a particular mating type may be relevant to study yeast mating. While we have previously used genetic modifications to label different mating types (e.g., https://pubmed.ncbi.nlm.nih.gov/24967739/), our strategy may facilitate this process by not requiring to genetically manipulate these cells and may as well be used to study other yeasts (e.g. Candida glabrataSte2p receptor for instance, recognizes the S. cerevisiae pheromone and consequently could be targeted by our peptides).
To clarify this, we have added the following sentence in the Introduction:
In this study we focus on testing the specificity of our designs and did not study the mechanism of internalization, because the mechanism of internalization should depend on the CPP used and the concentration of that peptide.
3) Figure 2. The figures are generally "stretched in the manuscript". Figure 2 is hard to read and the black line formed but all the data points not informative. The panel on the left had a max of 20000, the two panels on the right have a max of 200000. I am not sure if this is correct and if so, why the scales are different.
We appreciate the note. Indeed, the image was stretched. We have corrected this in this new version. It is also true that there is a difference in the Y-axis values, corresponding with the intensity detected for each of the cells analyzed using 3 different peptides. Hence, the 2 peptides reported on the right side of the image were detected with larger intensity than the first one on the left side of the image. These differences are reported in the main text of our work:
We observed that MATa cells exposed to the α-NLS-C peptide presented more selectivity towards MATα cells than NLS-α-CE and chimera peptides. All three peptides were preferentially internalized by MATa than by MATα cells. In summary, the three designed peptides showed selectivity for cells that expressed the receptor for the ligand embedded within the moonlighting CPP.
4) it is unclear why the peptides have lower-case letters, as this convention is usually used for D-amino acids.
We appreciate the note. We are now changing to upper case the amino acids letters to avoid confusion.
5) The diffuse fluorescence in Figure 7 is interpreted as "cytoplasmic". On one hand, it is possible. On the other hand, these images are not sufficient to make this claim. In particular, TAT peptides typically show nucleolar staining upon reaching the cytoplasm/nuclei. The red fluorescence observed here could instead be out of focus cell sticking. Moreover, diffuse fluorescence can also be observed as an imaging artifact (e.g. photolysis of endosomes by the excitation light of TMR during imaging). Finally, fixing cells can also lead to endosomal escape and a diffuse signal (it is not clear to me if/how/when the 4% paraformaldehyde step is use in the imaging of Figure 7).
We agree with the reviewer; there are many possibilities as to where exactly this peptide/fluorescence may be located. To clarify this, we changed the text referring to this figure as follows:
Interestingly, three different patterns of fluorescence were observed (see Figure 7): i) diffused fluorescence (blue arrow), ii) punctuated fluorescence (yellow arrow) and iii) a combination of these two (white arrow). TatNep is more often seen as a punctuated fluorescence while TatNepNoCPP and D-TatNepNoCPP as a diffused layer covering the cells.
6) the authors need to show that the signal they measure is not cell sticking (given the precipitation concerns). If it is in endosomes, they should demonstrate that. The internalization assay also needs to assess that the cells used for quantification are not dead/dying. A final concern is that the TAMRA signal could be partially quenched in some conditions but not otehrs, hence biasing the results. Fluorescence measurements from cell lysates (where quenching would be less likely) should be done and compared to those obtained by microscopy.
At this point of our work, we do not aim to determine the specific localization of these peptides. Instead, our goal is to show preferential internalization. The confocal microscopy experiments provide support to the internalization of these peptides.
On the other hand, it is true that toxic peptides may seem to be internalized, while instead cells are dead and consequently allow the passage of any molecules. The incubation time of the peptides was too short to expect to see any toxicity. In agreement with this, Figure 7 and supplementary material showed the morphology of cells, confirming the lack of toxicity, yet we did not provide any evidence of this lack of toxicity. To satisfy this point, we now show the results of toxicity measured for these peptides on mammalian cells as requested as supplementary material (Supplementary Figure S3). We also add the following sentence when referring to Figure 7:
No toxicity was observed during these experiments nor during an experimental quantification of toxicity (See Supplemental Figure S3), hence supporting the notion that these peptides are internalized through the Tat peptide once it is activated.
While the use of cell lysates would reduce the chance for TAMRA quenching, so this would happen after fixing cells with paraformaldehyde at 4% as reported in our study. Furthermore, the peptides used as controls further confirm the expected differences in internalization. It would be very unlikely that the quenching would preferentially act in the control peptides.
Reviewer 2 Report
In this paper, the authors describe the application for specific cellular uptake and reporting by CCP. The study is interesting. However, it is not clear how the CCP become specific for penetrating certain cell types. This hypothesis should be clarified for the reader, as now, this is not the case. Just adding a ligand to target a specific cell will not implicate this will increase specificity. In particular, as an expert in pharmacokinetics, biodistribution data will maybe show the opposite. Changing the charge or lipophilicity of small compounds may alter plasma protein binding capacities. How specific are these CCP as no kD was calculated. Also, for the lower concentrations in fig 6B, the CCP shows no intense fluorescence, which is a serious problem for in vivo targeting.
CPP are passing through biological membranes. However, the peptides used for this study (Table 1) show a high number of cationic residues. To my knowledge, this will cause non-specific uptake in bacteria, fungi, cancer cells and apoptotic cells. Has this non-specific behavior been evaluated as well?
Regarding the proteolytic breakdown of CCR using D and L enantiomers, this an interesting option to discuss.
Thus, as this paper is intentioned to reach pharmaceutics readers, the translational aspect must be incorporated as well. Another aspect is for increasing the mammalian cell-penetrating capacity, which will limit CCPs to pass endothelial barriers to reach the target cells behind. For yeasts, the purpose for increased specificity is not clear.
Author Response
In this paper, the authors describe the application for specific cellular uptake and reporting by CCP. The study is interesting. However, it is not clear how the CCP become specific for penetrating certain cell types. This hypothesis should be clarified for the reader, as now, this is not the case. Just adding a ligand to target a specific cell will not implicate this will increase specificity. In particular, as an expert in pharmacokinetics, biodistribution data will maybe show the opposite. Changing the charge or lipophilicity of small compounds may alter plasma protein binding capacities. How specific are these CCP as no kD was calculated. Also, for the lower concentrations in fig 6B, the CCP shows no intense fluorescence, which is a serious problem for in vivo targeting.
These are valid concerns, and we appreciate these comments. Indeed the biodistribution of drugs in an organism may depend on multiple factors, yet including a ligand that recognizes a receptor on a cell surface has been shown by different groups to effectively target the drug action at the animal level (see for instance these works: https://pubmed.ncbi.nlm.nih.gov/10470080/; https://pubmed.ncbi.nlm.nih.gov/22072637/). The same has been observed at the cellular level (see for instance these published works: https://pubmed.ncbi.nlm.nih.gov/12750379/;https://pubs.rsc.org/en/content/articlehtml/2020/cb/d0cb00114g). Furthermore, the use of selective flourescent CPP is not only relevant for animal studies. These peptides may be valuable tools for in vitro studies using cells. For instance, to be able to identify cells expressing a particular cell-surface marker may be useful in immunology studies, or in mating events associated with fungi. This last one is one of the motivations if using yeast cells in this study; we have added a sentence in the Introduction to clarify this aspect of our work.
Our hypothesis is that selectivity of CPP may be achieved by designing moonlighting peptides, that is, by embedding the CPP into a sequence that reduces the CPP-activity, hence requieres activation or by embedding a ligand that recognizes a cell-surface receptor into a CPP sequence. The advantages of using such designs are multiple: i) to reduce cost, ii) reduce the chances to be blocked by the immune system and iii) reduce the chance that the embedded peptide sequences will block each other. The basis of this hypothesis are explained in the introduction of our work.
The observation of the reviewer is correct, figure 6B shows no internalization at small concentration, specially for the control peptides, as expected. For instance, D-TatNepNoCPP was a control peptide that should not be activatable and consequently should not be internalized. Accordingly, figure 6B shows no fluorescent intensity for this peptide, specially at the lower concentrations tested. In comparison, the active TatNep and activatable TatNepNoCPP peptides were internalized in all concentrations, and their internalization was concentration dependent: at lower concentration, less fluorescent intensity was observed.
CPP are passing through biological membranes. However, the peptides used for this study (Table 1) show a high number of cationic residues. To my knowledge, this will cause non-specific uptake in bacteria, fungi, cancer cells and apoptotic cells. Has this non-specific behavior been evaluated as well?
Indeed, cationic peptides are able to penetrate many different cell types in a non-specific matter; the goal of our work is to show a novel approach to provide specificity to these cationic non-specific CPP. Our work uses a computer-based approach to add residues to either CPP or peptide ligands, to generate peptide sequences without CPP activity (activation design) or to create a CPP activity (targeting design), respectively. These computer-based designs allowed us to generate shorter peptide sequences than previous works where CPP activation or targeting have been reported. The results presented in this work confirm that our designs actually rendered targeted CPP on both yeast and mammalian cells.
Thus, as this paper is intentioned to reach pharmaceutics readers, the translational aspect must be incorporated as well. Another aspect is for increasing the mammalian cell-penetrating capacity, which will limit CCPs to pass endothelial barriers to reach the target cells behind. For yeasts, the purpose for increased specificity is not clear.
As noted before, the aim of our work at this stage is not to show the translational relevance of our findings, since these targeted fluorescent CPP may be used not only as drugs, but also as basic tools to study different cellular events. To further clarify this, we have also added the following sentence at the Discussion section:
Targeted CPP may have multiple applications. For instance, these peptides may be used to target drug action into specific cell types in cell cultures [31, 32] or in animals [10, 33]. Multiple studies aimed at studying cell physiology would benefit from having access to such peptides; for instance studies on yeast mating [34] or neuroscience [35]. We expect that our work may inspire other groups to use our computer-based designs in future studies.
Reviewer 3 Report
Morán-Torres et al. seeks to publish a study that used a computational approach to design moonlighting CPPs by adding 4 or up to 12 residues and examined their cell penetrating activities. The study is interesting, but it lacks adequate experimental and analytical evidence, which may not support their conclusions. Authors should do more experiments (to have sufficient n value) and analyze the data with appropriate statistical method to interpret their findings and thereby draw conclusions. Specific comments are below.
- Were the synthesized peptides linear? Did authors refold the synthesized peptides? How did authors confirm/verify the structure of the synthesized peptides? How did authors verify the addition of sequences to the N/C terminus of peptides? How about the structure of the peptides after modification of N/C terminus?
- What criteria were considered for selection of cell lines/genotypes?
- Did authors establish all the protocols in this study or they used previously established protocols? Please clarify and provide references appropriately if/where previously published methods were used. The entire method section should be written with appropriate references of previously published research, and/or mentioning with the modifications if authors have made any in the protocol.
- How did authors choose the concentration of penetrating peptide at 200 mM? Did they do any dose-response experiment? Authors should do dose-response or explain how they selected this concentration for use.
- Similarly, how did authors select the concentration 300 μg/ml Zeocin? It needs to be justified.
- Since authors dissolved peptides in DMSO/Water, did authors perform water/DMSO control experiments? If not, authors should to do these control experiments.
- Regarding internalization assays, authors first used peptides at a concentration of 10, 5 and 1 μM with cell lines for epifluorescent microscopy. Then again as mentioned, cells were incubated with either of the three peptides at a concentration of 5 μM for confocal microscopy. These are not quite clear. Please clarify and justify the use of the concentrations of the peptides.
- Number of experiments is not mentioned in the methods, nor in the figure legends. Only mentioned in legend for Figure-5, n=1. This is far from enough to make any assumption/conclusion. Authors should more experiments (n= ~5) for all experiments and mention the n value appropriately in all experimental figure legends.
- Authors should add a separate section as “Data analysis” in methods, with clear statement about what statistical analyses were used for what set of data, and why. Why is non-parametric Kruskal-Wallis test used? What is the number of experiments, why not parametric? Authors need to clarify.
- Authors should provide the name and information about the chemicals, reagents and solutions they used in the method section.
Author Response
Morán-Torres et al. seeks to publish a study that used a computational approach to design moonlighting CPPs by adding 4 or up to 12 residues and examined their cell penetrating activities. The study is interesting, but it lacks adequate experimental and analytical evidence, which may not support their conclusions. Authors should do more experiments (to have sufficient n value) and analyze the data with appropriate statistical method to interpret their findings and thereby draw conclusions. Specific comments are below.
Were the synthesized peptides linear? Did authors refold the synthesized peptides? How did authors confirm/verify the structure of the synthesized peptides? How did authors verify the addition of sequences to the N/C terminus of peptides? How about the structure of the peptides after modification of N/C terminus?
It is correct; all the peptides described in our study are linear, so there was no need to refold the synthesized peptides. We missed to specify this, so we have added the following sentence at the Methods section:
Please note all the peptides in this study are linear.
All peptides were chemically synthesized and verify by the peptide mass. This detail was not clarified in Methods, so we have added the following sentence to clarify this:
All these peptides were chemically synthesized by GenScript and purified >95%; the peptide sequences were validated by the peptide mass.
We did not check the secondary structure of these peptides because the designs were aimed to generate short peptide sequences. The idea we aimed to test was the selectivity of these peptides, so our work was focused on characterizing this aspect.
What criteria were considered for selection of cell lines/genotypes?
This is an important observation. For yeast cells, we did not originally specify the motivation to use these cells; we have now added some text at the Introduction and Discussion sections of our work to clarify this aspect.
In the case of mammalian cells, we have described that “To study the selectivity of the activatable cell penetrating peptides (ACPPs), we used HEK293T due to its extremely low level of Neprilysin (NEP) transcripts...”. NEP is a protease, hence it can cleave off a portion of the TatNepNoCPP peptide and activate the CPP activity as summarized in Figure 1.
Did authors establish all the protocols in this study or they used previously established protocols? Please clarify and provide references appropriately if/where previously published methods were used. The entire method section should be written with appropriate references of previously published research, and/or mentioning with the modifications if authors have made any in the protocol.
We appreciate the note. In the new version we report the previous works that we based our protocols on.
How did authors choose the concentration of penetrating peptide at 200 mM? Did they do any dose-response experiment? Authors should do dose-response or explain how they selected this concentration for use.
Before running the experiments with the AMNIS flux cytometer multiphotonic microscope, we had to run several tests to identify the appropriate peptide concentration required to identify in this equipment the correct amount of fluorescence signal. The reported concentration was the best concentration for the purpose of these studies. To clarify this, we have now added the following sentence:
Please note that the reported experimental conditions were selected among several trials on the AMNIs equipment, to identify the best conditions for these experiments.
Similarly, how did authors select the concentration 300 μg/ml Zeocin? It needs to be justified.
We added the following sentence in Methods to clarify this point:
The selected resistant cells were cultured under the Zeocin selection medium for approximately 2 months, then the HEK293T-NEP cells were maintained in the culture medium containing 300 µg/ml Zeocin; this concentration allowed the growth of the transfected cells, but not the wild-type...
Since authors dissolved peptides in DMSO/Water, did authors perform water/DMSO control experiments? If not, authors should to do these control experiments.
The DMSO concentration used was not toxic for the cells. To clarify this, we have now included as supplementary material (Supplementary Figure S3) cell viability results for all peptides and cell tested in this study.
Regarding internalization assays, authors first used peptides at a concentration of 10, 5 and 1 μM with cell lines for epifluorescent microscopy. Then again as mentioned, cells were incubated with either of the three peptides at a concentration of 5 μM for confocal microscopy. These are not quite clear. Please clarify and justify the use of the concentrations of the peptides.
To clarify this point, we added in Methods the following sentence:
Cells were incubated with either of the three peptides at a concentration of 5μM for 30 min (the fluorescence was strong and the precipitation reduced)...
Number of experiments is not mentioned in the methods, nor in the figure legends. Only mentioned in legend for Figure-5, n=1. This is far from enough to make any assumption/conclusion. Authors should more experiments (n= ~5) for all experiments and mention the n value appropriately in all experimental figure legends.
We appreciate this note. We have now added in the figure legends the number of measurements associated with each of the results. Many of our experiments report analysis on hundred to thousands of single cells, and each experiment was repeated at least 2 (only experiments reported in Figures 4A, 4C and 5) or 3 (all other experiments) times to validate the reported results. Thus, the n of our results are bigger than 5.
Authors should add a separate section as “Data analysis” in methods, with clear statement about what statistical analyses were used for what set of data, and why. Why is non-parametric Kruskal-Wallis test used? What is the number of experiments, why not parametric? Authors need to clarify.
We appreciate the note. We have added a section to clarify this point.
2.4.6 Data analysis
For the mammalian cells images obtained from epifluorescence microscopy, a normality Shapiro-Wilk test was done for each individual condition (27 conditions). None of the conditions showed a normal distribution (alpha = 0.05). To identify statistically equal distribution of peptide internalization a non-parametric Kruskal-Wallis test was implemented; pairs of experiments with p<0.05 were considered significantly equal.
Authors should provide the name and information about the chemicals, reagents and solutions they used in the method section.
We appreciate the note; we have added this information throughout the entire Methods section.
Round 2
Reviewer 2 Report
The authors didn't reply to my remark about calculations of the kD. They are essential to measuring the specific affinity for the cells.
Author Response
The authors didn't reply to my remark about calculations of the kD. They are essential to measuring the specific affinity for the cells.
The peptide sequences that have affinity for Neprylisin (Km 0.9 mM) or the alpha factor (Kd: 2-10 nM) are now included in our work. Please note that the tested concentrations in our experiments were above these values to ensure the reaction.
Reviewer 3 Report
Authors did not fully address this reviewer’s concerns.
The concerns that still need to be addressed are below-
- How did authors verify N/C terminal modification of peptides? Please clarify.
If authors used only linear peptides but not refolded peptides, how can they make sure the linear peptides are active in experiments?
- How did authors choose the concentration of penetrating peptide at 200 mM? Did they do any dose-response experiment?
Authors just added a sentence” Please note that the reported experimental conditions were selected among several trials on the AMNIs equipment, to identify the best conditions for these experiments”.
Authors should perform the dose-response and show the data in the manuscript.
- Similarly, how did authors select the concentration 300 μg/ml Zeocin? It needs to be justified.
Authors should perform the dose-response and show the data in the manuscript.
- Since authors dissolved peptides in DMSO/Water, did authors perform water/DMSO control experiments? If not, authors should to do these control experiments.
Supplementary Figure S3 does not show any DMSO control data. Authors should do these control experiments and show the data that DMSO does not have any toxic effects.
- Figure 5 legend still shows, n = 1. Make sure experiments are performed at least 3 times and mentioned properly.
Author Response
-
How did authors verify N/C terminal modification of peptides? Please clarify. If authors used only linear peptides but not refolded peptides, how can they make sure the linear peptides are active in experiments?
For each set of experiments, we used control peptides to validate the activity of the designed peptides. For instance, in the activation mechanism, we synthesized a peptide in D configuration that should not be activated by the cleavage of the Neprilysin protease (negative control). We also synthesized the peptide that should result from the Neprilysin cleavage (positive control). The designed peptide should be activated in cells expressing Neprilysin upon cleavage by this protease. We showed that such is the case (see Figures 6 and 7). In the case of the targeting mechanism, all peptides included the ligand that will be recognized by the alpha-pheromone receptor, a receptor expressed by the yeast MatA cells. To validate the specificity of the targeting, we used cells that do not express the alpha-pheromone receptor: the MatAlpha cells. We have also mixed MatA and MatAlpha cells together and expose this mixture of cells to the peptides; we showed that these peptides are preferentially internalized in MatA cells (see Figures 2, 3 and 4).
-
How did authors choose the concentration of penetrating peptide at 200 mM? Did they do any dose-response experiment? Authors just added a sentence” Please note that the reported experimental conditions were selected among several trials on the AMNIs equipment, to identify the best conditions for these experiments”. Authors should perform the dose-response and show the data in the manuscript.
The test for selectivity of these peptides was based on the presence or absence of the receptor recognized by any of these peptides. Figures 2, 3 and 4 show that all tested peptides are preferentially internalized by cells expressing the receptor recognized by the ligand embedded in the tested peptides. We understand that the use of a dose-response experiment is to show selectivity in the activity of the peptides, and that is what we are showing in these experiments.
What we tuned in the AMNIS equipment (flux cytometer coupled to a multiphotonic microscope) is the amount of fluorescence that allowed us to detect a significant number of fluorescent cells. An important aspect of these experiments is that the amount of cells used was very dense to allow for a fast detection of the fluorescent cells; we took pictures of cells every 5 minutes during the first 60 minutes of cells exposed to each of these peptides. As detailed in Methods, we used a cell density of 1.0 at 600 nm; for such large amount of cells we anticipated larger amounts of peptides needed to react with that many number of cells.
We hope this helps to clarify the idea behind the protocol and why the presented experiments address the concern noted by this reviewer.
-
Similarly, how did authors select the concentration 300 μg/ml Zeocin? It needs to be justified. Authors should perform the dose-response and show the data in the manuscript.
Zeocin is an antibiotic used to select transfected cells that express Neprelysin. This antibiotic was not part of the experiments reporting the internalization of the peptides, therefor it is not relevant for the aim of our work to perform a dose-response curve in this case. To clarify this aspect, we added the following phrase in Methods:
“Please note that Zeocin was not included in the internalization experiments reported in this work.”
-
Since authors dissolved peptides in DMSO/Water, did authors perform water/DMSO control experiments? If not, authors should to do these control experiments. Supplementary Figure S3 does not show any DMSO control data. Authors should do these control experiments and show the data that DMSO does not have any toxic effects.
We appreciate this note. Indeed the controls did not include DMSO. We want to clarify that the DMSO was only used to solubilize the peptides in a proportion of 1:4, 1 part of DMSO and 4 parts of total volume of water. This stock solution of peptide then was added to the media where the cells were growing, but the added amount of this stock solution was small. We did not specify this in our methodology and we believe that is the source of this comments from the reviewer. We are now specifying this aspect to clarify this problem. Additionally, we performed an experiment using that amount of DMSO used in peptides and it is now part of supplementary Figure S3 (see new Figures S3D and S3E). As expected, the amount of DMSO used to dissolve the peptides was not toxic to the cells. We have updated the methods description to account for these aspects of our work.
-
Figure 5 legend still shows, n = 1. Make sure experiments are performed at least 3 times and mentioned properly.
The motivation of figure 5 is to support the results presented in Figure 6 and 7; that is, we noted that some cell lines were more effective to internalize peptides. We assumed this could be explained by the amount of Neprilysin expressed by these cells. Figure 5 confirms the results reported in Figures 6 and 7. We are now including the results derived from a total of 6 westerns for HEK293T-NEP and 10 westerns for HeLa or HEK293T cells in Figure 5. We also updated the corresponding methods and figure legend to reflect these new data and corresponding statistical analysis.
Round 3
Reviewer 3 Report
Authors addressed most of the concerns, however, a couple of issues as mentioned below, should be done.
- Authors should be very specific about the number of experiments done. For example, for DMSO, authors stated in their response “Additionally, we performed an experiment using that amount of DMSO used in peptides…..”, without specifying n value in the method section, while figures S3D and S3E show bar graphs with error bars. Authors should write figure legends for all supplemental figures with specific number of experiments (n value) mentioned.
- Section 2.4.5, line 206. Please correct the word “essays” to “assays”.
Author Response
-
Authors should be very specific about the number of experiments done. For example, for DMSO, authors stated in their response “Additionally, we performed an experiment using that amount of DMSO used in peptides…..”, without specifying n value in the method section, while figures S3D and S3E show bar graphs with error bars. Authors should write figure legends for all supplemental figures with specific number of experiments (n value) mentioned.
We appreciate the note. Indeed, we missed to describe that aspect of our work. As suggested, we have added a figure legend stating the number of data points presented in each plot. The figure legend now reads:
Three independent biological experiments were conducted for each condition and each period of time, three replicates were used in all cases. The plot presents the average of 9 data points and the bars show the mean +/- standard deviation.
-
Section 2.4.5, line 206. Please correct the word “essays” to “assays”.
Thanks for the note. We changed the word as indicated.